# D^2^BGAN: Dual Discriminator Bayesian Generative Adversarial Network for Deformable MR–Ultrasound Registration Applied to Brain Shift Compensation

**DOI:** 10.3390/diagnostics14131319

**Published:** 2024-06-21

**Authors:** Mahdiyeh Rahmani, Hadis Moghaddasi, Ahmad Pour-Rashidi, Alireza Ahmadian, Ebrahim Najafzadeh, Parastoo Farnia

**Affiliations:** 1Department of Medical Physics and Biomedical Engineering, Tehran University of Medical Sciences (TUMS), Tehran 1461884513, Iran; mrahmani@razi.tums.ac.ir (M.R.); h-moghaddasii@razi.tums.ac.ir (H.M.); ahmadian@sina.tums.ac.ir (A.A.); 2Research Center for Biomedical Technologies and Robotics (RCBTR), Advanced Medical Technologies and Equipment Institute (AMTEI), Imam Khomeini Hospital Complex, Tehran University of Medical Sciences (TUMS), Tehran 1419733141, Iran; 3Department of Neurosurgery, Sina Hospital, School of Medicine, Tehran University of Medical Sciences (TUMS), Tehran 11367469111, Iran; apourrashidi@sina.tums.ac.ir; 4Department of Medical Physics, School of Medicine, Iran University of Medical Sciences, Tehran 1417466191, Iran; 5Department of Molecular Imaging, Faculty of Advanced Technologies in Medicine, Iran University of Medical Sciences, Tehran 1449614535, Iran

**Keywords:** neurosurgery, brain shift, intraoperative ultrasound, deep learning, generative adversarial network

## Abstract

During neurosurgical procedures, the neuro-navigation system’s accuracy is affected by the brain shift phenomenon. One popular strategy is to compensate for brain shift using intraoperative ultrasound (iUS) registration with pre-operative magnetic resonance (MR) scans. This requires a satisfactory multimodal image registration method, which is challenging due to the low image quality of ultrasound and the unpredictable nature of brain deformation during surgery. In this paper, we propose an automatic unsupervised end-to-end MR–iUS registration approach named the Dual Discriminator Bayesian Generative Adversarial Network (D^2^BGAN). The proposed network consists of two discriminators and a generator optimized by a Bayesian loss function to improve the functionality of the generator, and we add a mutual information loss function to the discriminator for similarity measurements. Extensive validation was performed on the RESECT and BITE datasets, where the mean target registration error (mTRE) of MR–iUS registration using D^2^BGAN was determined to be 0.75 ± 0.3 mm. The D^2^BGAN illustrated a clear advantage by achieving an 85% improvement in the mTRE over the initial error. Moreover, the results confirmed that the proposed Bayesian loss function, rather than the typical loss function, improved the accuracy of MR–iUS registration by 23%. The improvement in registration accuracy was further enhanced by the preservation of the intensity and anatomical information of the input images.

## 1. Introduction

The improvement of the survival rates of patients, particularly in neurosurgery, depends heavily on the maximal safe resection of the tumor. However, achieving the maximal safe resection of brain tumors while preserving the normal cerebral tissue is challenging due to the resemblance of tumors to the brain parenchyma and the presence of the adjacent vital structures [1,2]. In this regard, neuro-navigation technologies have demonstrated significant potential in accurately localizing tumors and vital brain structures during minimally invasive surgical procedures. These systems provide accurate localization and guidance during surgical procedures by updating pre-operative images, such as magnetic resonance (MR) and computed tomography (CT) images. This is performed by the registration of the patient’s coordinates during surgery, enabled by the IR camera with pre-operative images [3]. The navigation accuracy is influenced by multiple factors, including the pre-operative imaging quality and resolution, technical aspects like instrument calibration and the tracking device’s precision, and the accuracy of the registration algorithms. Meanwhile, the accuracy of neuro-navigation systems can also be limited by brain shift, which refers to the deformation and displacement of the brain tissue during the operation [4]. A shift in brain tissue causes an anatomical discrepancy between the pre-operatively acquired data and the actual surgery field [5].

This phenomenon is influenced by various physical, surgical, and biological factors in both the cortical and deep regions of the brain. Meanwhile, intraoperative imaging modalities such as X-ray, computed tomography (CT), magnetic resonance imaging (MRI), fluorescence imaging, and ultrasound (US) could provide accurate anatomical localization during surgery and have the potential to compensate for brain shift. However, each of these methods has its well-known advantages and limitations. The radiation exposure and low spatial resolution in CT, the requirement for an MR-compatible operating room with expensive equipment and the time-consuming nature of MRI, and the limited imaging depth in fluorescence imaging are the major challenges of the most common intraoperative imaging modalities [6].

Among these modalities, ultrasound is highly regarded for its real-time imaging capabilities, availability, cost-effectiveness, and non-ionizing nature. Intraoperative ultrasound (iUS) is considered a promising imaging method for the delineation of tumor margins and normal brain tissue [7]. The iUS can compensate for brain shift during neurosurgery and update patient coordination by registering the iUS with pre-operative MR images. The quality of MR-iUS registration during surgery is limited due to the low contrast and quality of ultrasound images, which are operator-dependent, and the complex and unpredictable nature of brain shift [8].

Numerous investigations have attempted to address the challenges associated with multimodal image registration methods for brain shift compensation. These methods include feature-based [9] and intensity-based [10] approaches, in which the latter has demonstrated greater success [6]. Intensity-based methods compare the intensity patterns in images via correlation metrics, and feature-based methods rely on the precision of feature selection. For this purpose, Rivas et al. [11] introduced an innovative algorithm named MR–iUS Registration via Patch-based Correlation Ratio (RaPTOR), which calculates the local correlation ratio. Farnia et al. [12,13], focusing on enhancing the residual complexity value in the wavelet and curvelet domains, proposed hybrid techniques aimed at aligning the echogenic structures in MR with iUS images. Zhou and Rivaz [14] presented a non-rigid symmetric registration approach based on ultrasound images captured before and after tumor resection, addressing challenges in evaluating typical post-resection images.

Riva et al. [15] conducted a comprehensive evaluation of brain shift compensation at three critical time points during surgery. The initial assessment occurred before the dural opening, employing a linear correlation of linear combination. Subsequent evaluations after dural opening involved normalized cross-correlation registration and deformable B-spline registration methods. Finally, after the resection of the brain tumor, iUS was applied to the pre-operative image. In another study, Wein proposed a model based on the LC2 multimodal similarity measure [16], offering another perspective on brain shift compensation. Additionally, Drobny et al. [17] presented a block-matching approach utilizing NiftyReg [18] for the automatic registration of MR and iUS images.

In CuRIOUS 2018 [19], attempts were made to refine MR–iUS registration algorithms, such as the Deformable Registration via Attribute Matching and Mutual-Saliency Weighting (DRAMMS) algorithm proposed by Machado et al. [20] and Deeds/SSC, which comprises linear and non-rigid registration, implemented by Heinrich et al. [21]. Despite the advancements in the registration accuracy achieved by these algorithms, their utilization has resulted in the increased computational complexity of the registration process, which is not suitable for the real-time compensation of brain shift. On the other hand, deep learning (DL) has recently shown great potential and gained attention in medical applications due to its superior performance in a variety of image processing applications, such as object detection, feature extraction, image segmentation, etc. For the first time, DL approaches were used for brain shift correction in CuRIOUS 2018. Sun et al. [22] presented a DL framework for non-rigid MR-iUS registration using a 3D convolutional neural network (CNN) consisting of a feature extractor, a deformation field generator, and a spatial sampler. As an imitation game, Zhong et al. [23] suggested a learning-based strategy for intraoperative brain shift correction. They trained a neural network to imitate the demonstrator’s behavior and anticipate the optimal deformation vector. Moreover, Zeineldin et al. [24] employed a 3D CNN for deformable MR–iUS image registration. Despite their potential for real-time compensation, these DL approaches have faced challenges in achieving high accuracy.

Automated DL approaches for MR–iUS registration reduce the registration time by learning a global function for multimodal, non-rigid image registration. These DL approaches can be efficiently applied to a pair of test images, saving substantial time during the registration procedure. Therefore, all of these methods are suitable for the real-time compensation of brain shift, but they have not achieved adequate image registration error rates. Consequently, achieving high accuracy along with quick registration is a major issue that remains challenging for brain shift compensation during neurosurgery. On the other hand, preparing ground truth data requires a lot of experience and is time-consuming. Furthermore, the non-rigid and unpredictable nature of brain shift has led to unsupervised learning, in which the challenges associated with the ground truth data generation and optimization methods would be eliminated.

To address the multimodal image registration challenges with the benefit of DL methods, in this study, for the first time, we propose an unsupervised end-to-end registration method via the Dual Discriminator Bayesian Generative Adversarial Network (D^2^BGAN). This network consists of two types of neural networks based on deep convolutional generative adversarial networks (GAN), including a generator, two discriminators, and dual discriminator GAN [25]. We optimize the architecture of the dual discriminator GAN by introducing a Bayesian loss function to improve the generator’s functionality and a mutual information loss function to the discriminator. To this end, we employ two public datasets, RESECT and BITE, acquired under various conditions, with one used to train the models and the other to evaluate the performance of the registration method and the data compatibility.

The subsequent sections discuss the adversarial training of the generative model in Section 2 and present the experimental results on labeled datasets in Section 3. Finally, the paper culminates in Section 4 and Section 5, highlighting the discussion and conclusions.

## 2. Materials and Methods

In this section, we first explain the basic theory of GANs. Then, we discuss our proposed registration approach based on GANs and the proposed hybrid loss functions. Furthermore, a comprehensive explanation is provided regarding the utilized datasets in this study, followed by the data preparation steps. The end of this section discusses the details of our training and evaluation metrics.

### 2.1. Generative Adversarial Network (GAN)

The rise of GANs [26] has led to a new era in medical image processing. One notable aspect of a GAN is its ability to produce superior results without relying on labeled data. Generative artificial intelligence technologies such as GANs and diffusion models are rapidly evolving, showing great potential across various domains [27,28,29].

This is accomplished through the competitive interplay between the generator (G) and discriminator (D) networks. As a result, GANs are proving to be a cutting-edge tool, achieving satisfactory results in various medical applications, such as image registration [30].

GANs outperform other networks by permitting two models to be trained simultaneously. The GAN training process involves an adversarial learning setup where the G and D models play distinct roles. G is responsible for generating synthetic data samples that aim to resemble real data, while D acts as a discriminator to distinguish between real and generated samples. G tries to produce diverse samples that challenge D’s ability to accurately identify them as false. This dynamic interplay between G and D drives the GAN’s learning process to achieve improved performance.

### 2.2. D^2^BGAN Framework

The proposed procedure of the D^2^BGAN pipeline for MR–iUS registration with one generator, G, and two discriminators, D_1_ and D_2_, is shown in Figure 1a. During training, G is trained with the input images and generates images that not only appear realistic but also contain the details present in the input images. By incorporating these details, G aims to confuse D_1_ and D_2_, which are responsible for distinguishing between real and generated images. The training process is conducted to intentionally confuse D_1_ and D_2_ and ensure that the generated registered images closely resemble the input images in terms of their visual characteristics. In this network, two discriminators are used because of the significantly different natures of iUS and MRI images.

The generator is an autoencoder that consists of an encoder and decoder network, as illustrated in Figure 1b. The encoder and decoder of the generator include five convolutional layers with a 3 × 3 kernel size, as illustrated in Figure 1b. To avoid vanishing gradients and speed up the training process, batch normalization is applied to all convolutional layers, and the strides of all layers are set to 2. In addition, the ReLU activation function is used to accelerate convergence.

D_1_ and D_2_ are trained to distinguish between MR–iUS images and the registered image. Therefore, the discriminators constitute a two-channel layer having both the sampled data and the source image.

In the discriminators, all convolutional layers have a stride of 2, with a ReLU activation function and batch normalization. The last layer of the discriminators is a dense layer with a Tanh activation function. This final layer predicts the likelihood of the input image belonging to the source images, rather than being generated by the generator G. This setup enables the discriminators to provide feedback to the generator and guide its learning process based on the authenticity of the generated images.

In a dual discriminator structure, it is crucial to maintain a balance between the discriminators while considering the adversarial relationship with the generator. The strength or weakness of each discriminator affects the overall efficiency of the training process. To achieve a balance, the network design and training procedures ensure that both discriminators have the same architecture.

When the generator G fails to deceive the discriminator D and establishes an adversarial relationship, it may start generating images randomly, leading to a distortion in the generative model. To mitigate this, the training objective for G is defined by minimizing the adversarial loss, which can be expressed as follows:(1)$$\underset{G}{\min}\underset{D_{1},D_{2}}{\max}E\left[logD_{1}\left(iUS\right)\right]+E\left[\log\left(1-D_{v}\left(G\left(iUS,MR\right)\right)\right)\right]+E\left[\log D_{2}\left(MR\right)\right]+E[\log(1-D_{2}((iUS,MR)))]$$

Here, the discriminators’ goal is to maximize Equation (1). The divergence between the probability distributions of the generator G and the two real discriminators, D_1_ and D_2_, will become smaller. This adversarial training process creates a dynamic interplay between G and D, where G continually improves its ability to generate realistic images that can deceive D, while D simultaneously learns to better discriminate between real and generated images. Through this adversarial training, G gradually improves its generative capabilities and produces higher-quality images.

In the following, a measure of the distance between the probability distributions is used to define the G and D losses. The false data distribution can only be affected by G; therefore, during generator training, we omit the real data. Therefore, the G loss function, as an adversarial loss $L_{G}$, is defined as
(2)$$L_{G}=E\left[\log\left(1-D_{1}\left(\left(iUS,MR\right)\right)\right)\right]+E[\log(1-D_{2}\left(G(iUS,MR\right)))]$$

The discriminators differentiate between the source and moving images and the registered image. The Jensen–Shannon divergence between the distributions can be calculated using the adversarial losses of the discriminators, which can be used to determine whether the intensity or texture information is unrealistic. As a result, the matching of the realistic distribution is encouraged. The following formulas define the adversarial losses:(3)$$L_{D_{1}}=E\left[-\log D_{1}(iUS)\right]+E\left[-\log(1-D_{1}\left(G(iUS,MR\right)))\right]$$
(4)$$L_{D_{2}}=E\left[-\log D_{2}(MR)\right]+E\left[-\log(1-D_{2}\left(G(iUS,MR\right)))\right]$$

By minimizing these adversarial losses, G learns to generate images that are more likely to be classified as real by the discriminators. This encourages the generator to produce images that closely reflect the realistic distribution of the source and moving images.

To improve the network performance, two other loss functions, one for implementation in G and the other for D, are presented as follows.

**Mutual information loss function:** We aim to identify the interdependence between the source and fixed images and the registered image, which can be measured using mutual information (MI). MI refers to the quantity of information learned about the registered image from the other input images. Therefore, the $D_{1}\;\mathrm{and}\;D_{2}$ loss functions could be defined based on the MI of the registered and the input images, MR and iUS; therefore, we define the MI loss function for $D_{1}\;\mathrm{and}\;D_{2}$ as
(5)$$L_{{MI}_{D_{1}}}=1-\underset{\mathrm{registered}}{\min}\underset{\mathrm{US}}{\max}\sum_{i}^{M}\sum_{j}^{N}p\left({registered}_{i},{iUS}_{j}\right)log[p\left({registered}_{i},i{US}_{j}\right)/p\left({registered}_{i})p(i{US}_{j}\right)]$$
(6)$$L_{{MI}_{D_{2}}}=1-\underset{\mathrm{registered}}{\min}\underset{\mathrm{MRI}}{\max}\sum_{i}^{M}\sum_{j}^{N}p\left({registered}_{i},{MR}_{j}\right)log[p\left({registered}_{i},{MRI}_{j}\right)/p\left({registered}_{i})p({MRI}_{j}\right)]$$

The MI loss function serves as a regularization term for the discriminator model during both the training and image generation processes. By including the MI loss in the overall objective function, the discriminator is encouraged to learn and capture the mutual information between the registered image and the input images.

**Bayesian loss function:** Based on the prior knowledge of the MR and iUS images, Bayes’ theorem defines the likelihood of the registered image with the input images. It is a probabilistic framework that provides an interpretation of the probability, focusing on the state of knowledge rather than the intensity or similarity of the images. Bayes’ theorem is valuable in this context because it is not reliant on the similarity between the pixel intensities, making it efficient and informative for nonlinear problems where pixel intensity independence is assumed.

One advantage of Bayesian probability is the determined posterior distribution, which makes the computation simple. Bayesian approaches enable the prediction of the likelihood of the registered image, which is referred to as the predictive probability. The predictive distribution represents the probability distribution of the registered image based on the MR and iUS images. Therefore, based on Bayes’ theorem, we define the Bayes loss function for the generator as
(7)$$L_{B_{G}}=1-\underset{\mathrm{registered}}{\min}\underset{\mathrm{MRI}}{\max}\sum_{i}^{M}\sum_{j}^{N}p\left({MRI}_{j},i{US}_{j}|{registered}_{i}\right).p({registered}_{i})/p\left({MRI}_{j},{iUS}_{j}\right)$$
where p(registered image) and p(MRI,iUS) are the known prior probabilities, and $p\left(MRI,iUS|registered\;image\right)$ is known as the conditional probability, in which M and N are the numbers of pixels in the registered image and input images, respectively.

### 2.3. Dataset

In this study, experiments were conducted on the public RESECT [31] and BITE [32] datasets, which contain pre-operative MR and iUS images. The RESECT dataset contains data from 22 patients with low-grade glioma who received surgery at St. Olav’s University Hospital, Norway. Each patient was scanned using the Gadolinium-enhanced T1-weighted and T2-FLAIR MR protocols with the 1.5T Siemens Magnetom Avanto to reveal the anatomy and pathology, with a voxel size of 1 × 1 × 1 mm^3^ and a resolution of 256 × 256 × 192 pixels. The series of B-mode US images was obtained with a 12FLA-L linear probe with a voxel size of 0.14 × 0.14 × 0.14 mm^3^ to 0.24 × 0.24 × 0.24 mm^3^ before, during, and after tumor resection to track the surgical progress and tissue deformation. Overall, 338 corresponding anatomical landmarks were identified between MR and iUS. Moreover, 264 paired slices were extracted from the 3D volumes of MR and iUS.

The BITE dataset contains data from 14 patients with low-grade and high-grade glioma who received surgery at the Montreal Neurological Institute, Montreal, Canada. Each patient was scanned with the 1.5T General Electric Sigma EXCITE, with a voxel size of 1 × 1 × 1 mm^3^ and a resolution of 256 × 256 × 256 pixels. The series of B-mode US images was obtained with a phased-array transducer with a voxel size of 0.3 × 0.3 × 0.3 mm^3^. Overall, 355 corresponding anatomical landmarks were identified between MR and iUS.

### 2.4. Training Details

Throughout the training process, the principle is to create an adversarial relation between the generator and discriminators. The epoch is set at 200 to maintain the balance between the generator and discriminators. However, there are still instances where these networks have been trained but are unable to attain balancing conditions. Utilizing L_B_ for the generator and L_MI_ for the discriminators may help to prevent the above-mentioned issue. We validate the proposed approach on the RESECT and BITE datasets. In total, 264 slices from 22 cases of the RESECT dataset are extracted and utilized to train the networks, without any ground truth alignment. In this fully automated algorithm, no pre-processing steps and no manual initialization are required.

To test and validate the algorithm, 110 and 112 pairs of corresponding slices of iUS and MR images from the RESECT and BITE datasets are extracted, respectively. The entire network is trained with a learning rate of 1 × 10^−5^, exponentially decaying to 0.85 of the original value after each epoch. The batch size is set as 24. The network is trained on an Intel^®^ Core ™ i9-9900k CPU @ 3.60GHz with 32.0GB RAM and a GeForce RTX 3060Ti.

### 2.5. Evaluation Metric

We used the common evaluation metric of the mean target registration error (mTRE) for MR–iUS registration, similar to others in this field [33]. It is the average distance between corresponding landmarks in each pair of MR and iUS images. The mTRE is defined as
(8)$$mTRE=\frac{1}{N}\sum_{1}^{N}\left\|T\left(x_{i}\right)-x_{i}^{\prime }\right\|,\;i=1,2,\ldots ,N$$
where $x_{i}$ and $x_{i}^{\prime }$ represent the corresponding landmark locations annotated by an expert in the MR and iUS images, respectively, and T is the MR to iUS transform, calculated from the registration procedure.

## 3. Results

To evaluate the performance of the unsupervised end-to-end D^2^BGAN registration method for brain shift compensation, we compared the mTRE when identifying anatomical landmarks between MR and iUS in the registered images. Table 1 summarizes the mTREs of the pre- and post-registration stages of the proposed method for all 22 RESECT cases individually.

During the multimodal registration process, we utilized two different sets of loss functions, including *L*_1_ and *L*_2_. The *L*_1_ loss function was used *L_G_* for G, with $L_{D_{1}}$ and $L_{D_{2}}$ for *D*_1_ and *D*_2_, respectively. Furthermore, in the *L*_2_ loss function, we incorporated additional terms. For G, we used $L_{G}+L_{B_{G}}$, and we used $L_{D_{1}}+L_{{MI}_{D_{1}}}$ and $L_{D_{2}}+L_{{MI}_{D_{2}}}$ for the *D*_1_ and *D*_2_ discriminators, respectively.

The evaluation of the D^2^BGAN registration method on the RESECT dataset demonstrates a significant improvement in the registration accuracy. The initial error, which was measured at 5.42 mm, is reduced to 0.75 mm after applying the proposed method, with a statistically significant difference (*p*-value = 0.000016). This indicates a clear advantage of the D^2^BGAN registration method in terms of reducing the registration error on the RESECT dataset.

To compare the measurement of the TRE for the L_1_ and L_2_ loss functions, we analyze how the mTRE performs throughout registration. In Table 1, the *p*-values (0.000026 for the L_1_ and 0.000016 for the L_2_) indicate a highly significant reduction in the mTRE values, confirming that the improvements are not due to random variation and highlighting the effectiveness of the D^2^BGAN approach.

To further validate the generality of the proposed method, we also evaluated its performance on the BITE dataset. Table 2 provides a summary of the mTREs before and after registration using the D^2^BGAN method for all 14 cases in the BITE dataset. The individual mTRE values in Table 2 reflect the accuracy of landmark identification between the MR and iUS images pre- and post-registration.

Table 2 demonstrates the satisfactory performance of the D^2^BGAN method on the BITE dataset, which led to an improvement in the registration result of 1.35 ± 0.49 mm from the initial error of 4.18 ± 1.84 mm. By achieving reduced mTRE values on the BITE dataset as well, the D^2^BGAN registration method demonstrates its capability to improve the registration accuracy consistently across multiple datasets. Furthermore, Figure 2 presents distribution plots depicting the variations in the mTRE for the 22 cases in the RESECT dataset and the 14 cases in the BITE dataset. The *y*-axis shows a box plot with the mTRE for each combination (maximum, minimum, interquartile range, and median).

For a qualitative evaluation of the registration results on the RESECT dataset, visual examples of MR to iUS registration are provided in Figure 3. Specifically, cases 9, 15, and 18 of the dataset are demonstrated. In the last row of the figure, a color overlay of the iUS image over the pre-operative MR image is displayed, highlighting the improved alignment achieved through the registration process. These examples provide a visual representation of the effectiveness of the registration method in aligning the MR and iUS images and improving the spatial correspondence between them.

In addition to the evaluation of the BITE dataset, a qualitative assessment of the registration results on the RESECT dataset is also provided. Figure 4 provides visual examples of MR–iUS registration for cases 12, 13, and 14 of the dataset. The improved alignment between the two modalities is visually demonstrated through the yellow circles, which highlight the areas of improved correspondence.

Moreover, Figure 5 indicates the brain shift in the sagittal, axial, and coronal slices. The rows sequentially indicate pre-operative imaging MR, iUS, and registered images. The overlay between the iUS acquisition and the pre-operative MR is shown in the last row.

To compare our proposed method to others, Table 3 presents the mTRE for the proposed D^2^BGAN and other methodologies found in the literature on MR–iUS registration performed on the RESECT dataset. As a result, a comparison of our registration method and other MR-iUS registration approaches, such as LC^2^ [10], SSC/Deeds [21], NiftyReg [18], and cDRAMMS [34], as well as learning methods, FAX [23] and CNN [22], is achieved.

The results demonstrate that our proposed D^2^BGAN method achieves the lowest mTRE of 0.75 ± 0.30 mm, indicating superior registration accuracy compared to the other methods. LC^2^, SSC, FAX, and cDRAMMS also show relatively low mTRE values, while NiftyReg and CNN exhibit higher mTRE values. On the other hand, the results’ improvement is also due to the loss function used in the network training process. For the G, we used $L_{G}+L_{B_{G}},$ and we used $L_{D_{1}}+L_{{MI}_{D_{1}}}$ and $L_{D_{2}}+L_{{MI}_{D_{2}}}$ for the *D*_1_ and *D*_2_ discriminators, respectively. Figure 1 indicates the learning curves for the validation loss of the generator and both discriminators. The curve illustrates the progression of the loss functions over time, highlighting how each component’s performance evolves throughout the training process. As shown in Figure 6, the convergence process of network D_1_ is slower than that of network D_2_ due to the more complex nature of iUS images. These findings highlight the effectiveness of our proposed D^2^BGAN method in achieving highly accurate MR–iUS registration on the RESECT dataset.

## 4. Discussion

This paper introduced a DL-based MR–iUS registration approach for brain shift compensation during image-guided neurosurgery. We developed the D^2^BGAN model, which is trained in an unsupervised end-to-end manner. In the D^2^BGAN pipeline, a single generator and two discriminators are utilized, and experiments were run on the training images extracted from the RESECT dataset.

Two discriminators were used in our study to accommodate the diverse nature of intraoperative iUS and MRI images. Initially aimed to share information, this approach proved ineffective, increasing the complexity without aiding convergence. Assigning separate tasks to each discriminator improved the convergence, effectively managing the distinct characteristics of each modality. This approach aims to eliminate the dependence of image registration on the pixel intensity variation in different modalities so that the network can learn nonlinear transformations.

During training, the adversarial loss can cause some problems, exhibited by the result with more details in the registered image, and the generator loss can confuse the received features from the inputs. Therefore, the network was optimized by introducing a Bayesian loss function to improve the generator’s functionality. The Bayesian loss function can mitigate the dependence of image registration on the pixel intensity variation of different modalities, and using the mutual information loss function as a similarity measure enhances the network capabilities. In the image generation process, the MI loss function is utilized to guide the generation of high-quality images that maintain mutual information with the input images. By optimizing the generator based on the MI loss function, the generated images are more likely to preserve the relevant information from the input images, resulting in more realistic and visually consistent outputs.

The pre-registration accuracy is influenced by various factors, including the size and location of the craniotomy, the patient’s head position, the biological shift amount, and the location from which the physician performs the iUS imaging. As observed in Table 1, the pre-registration value can vary from 1.13 to 19.76 mm depending on the mentioned factors, which can significantly affect the accuracy of registration. One of the strong points of the proposed algorithm is its compensation for the occurred shift and its reduction to the range of 0.51 to 1.76, indicating the algorithm’s ability to compensate for the occurred biological shift, which would be different for each case. As shown in Table 1, a significant improvement is revealed in the results over the initial alignment. The D^2^BGAN network with the L_1_ loss function reduced the initial mTRE by about 81%, while, with the proposed L_2_ loss function, the mTRE reduction is about 86%. When changing the loss function from L_1_ to L_2_, in cases 5, 13, 16, and 27, the error rate did not change, but, in some cases, such as 6 and 23, we observed the greatest changes, about 65%, over the initial alignments. By combining these two loss functions, D^2^BGAN can address the issue of exhibition and clearer texture details and generate a high-quality registered image. It is important to note that our fully automated algorithm does not require any manual initialization or prior knowledge, which further enhances the efficiency and usability of this approach.

Additionally, an external test was performed to evaluate the data compatibility in the models. The output results of the MR–iUS registration for the BITE dataset were obtained and compared with the ground truths. The external test characterizes how the model can be utilized generally in varied data, which are common in clinical settings. As can be seen in Figure 4 and Table 2, although the network was not trained using the BITE dataset, the performance of D^2^BGAN on it showed an approximately 67% improvement compared to the initial alignments.

Moreover, the algorithm must be capable of registering different fields of view in multimodality registration. In Figure 5, the result of MR–iUS registration in the sagittal, axial, and coronal planes for case 4 of the RESECT dataset are illustrated. Furthermore, an experienced neurosurgeon evaluated our registration results and approved the aligned results of the presented algorithm.

Table 3 displays the initial and final landmark errors for the proposed D^2^BGAN network and other methodologies found in the MRI–iUS registration literature and performed on the RESECT dataset. The proposed method is compared to conventional methods, LC2 [10], SSC [21], NiftyReg [18], and cDRAMMS [34], as well as learning methods, FAX [23] and CNN [22]. As can be seen in Table 3, D^2^BGAN ranks first on the RESECT dataset, with an average mTRE of 0.75 ± 0.30 mm, followed by the learning-based method FAX, with an mTRE of 1.21 ± 0.55 mm. In comparison to NiftyReg and cDRAMMS, conventional techniques such as LC2 and SSC have lower mTRE values and indicate moderate registration accuracy. These approaches are straightforward and computationally efficient, but they cannot adequately reflect the delicate spatial relationships found in medical images because of their reliance on hand-crafted features and optimization strategies. On the other hand, learning-based methods, our proposed D^2^BGAN, and the FAX network demonstrated superior registration accuracy, which reveals that DL approaches have the potential to be used to resolve medical image registration challenges.

In particular, D^2^BGAN’s inclusion as a registration method significantly influenced the registration accuracy and provided the capability to preserve the intensity and anatomical information. The results show that D^2^BGAN can perform automatic and accurate MR–iUS image registration; therefore, it might be used in image-guided neurosurgical operations. The robustness of the network allows for a significant intensity- and scale-invariant transform in the registered image, demonstrating the effectiveness of its use for multimodal registration.

The limitations of the study primarily revolve around the subjective nature of the evaluation metrics and the lack of comprehensive validation techniques. Utilization on manually annotated landmark correspondences introduces human errors and may not adequately represent the errors in clinical targets, like tumor edges. Moreover, the absence of rigorous validation methods, such as exploring the Dice overlap on actual tumor segmentations and the properties of the deformation field, prevents the thorough assessment of the registration accuracy. These limitations highlight the need for more robust evaluation methods and metrics to ensure the accurate assessment of the registration accuracy in future studies.

## 5. Conclusions

Neuro-navigation technology provides guidance to find the optimal route to the target during neurosurgical procedures. However, neuro-navigation systems have been unreliable due to the brain shift that occurs during neurosurgery. To address this issue, MR–iUS registration is introduced for brain shift compensation. However, due to the unpredictable nature of brain deformation and the low quality of ultrasound images, finding a successful multimodal registration approach remains challenging in this area. Therefore, this study attempted to increase the accuracy of MR–iUS registration to identify intraoperative brain shift via D^2^BGAN.

D^2^BGAN’s inclusion as a registration method had a significant impact on the registration accuracy and provided the capability to preserve the intensity and anatomical information. The network’s performance enabled us to obtain an intensity- and scale-invariant transform to a great extent in the registered image, which demonstrates the efficacy of its use for multimodal registration. The results of the proposed method compared to other methods reveal that, in addition to real-time MR–iUS registration, this method also increases the registration accuracy. This encourages the further development of DL-based approaches for multimodal, non-rigid registration.

## Figures and Tables

**Figure 1 diagnostics-14-01319-f001:**
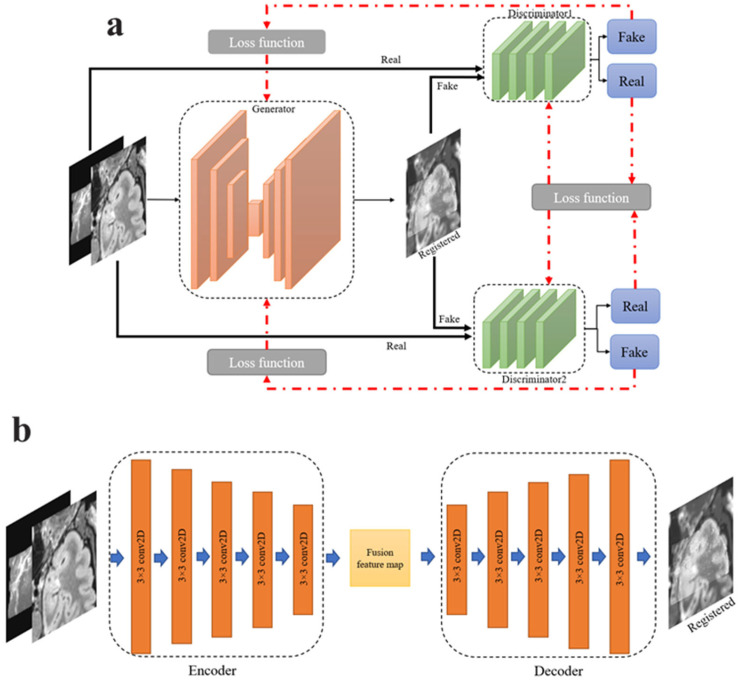
The D2BGAN pipeline with two discriminators and one generator. (**a**) The D^2^BGAN framework; (**b**) the generator network: the architecture of the generator network with a 3 × 3 filter size. Conv2D is a convolutional layer that obtains k feature maps, with batch normalization and a ReLU activation function.

**Figure 2 diagnostics-14-01319-f002:**
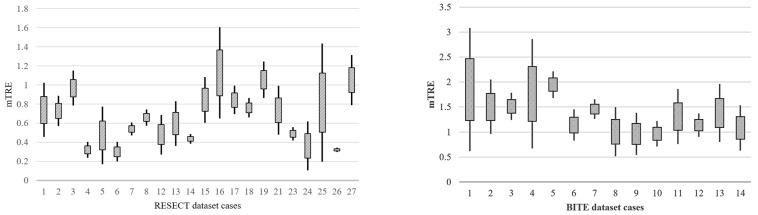
Distribution plots of mTRE for 22 cases of the RESECT dataset and 14 cases of the BITE dataset.

**Figure 3 diagnostics-14-01319-f003:**
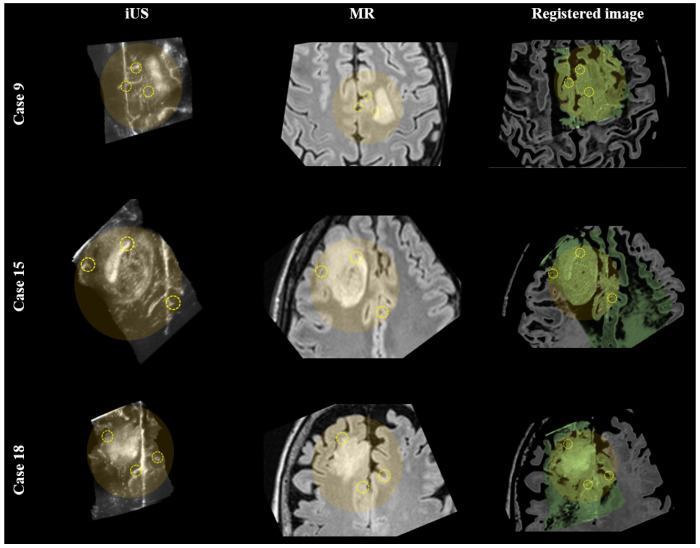
MR to iUS registration results. The first, second, and third rows correspond to cases 9, 15, and 18 of the RESECT dataset, respectively. The left, middle, and right columns correspond to the iUS and pre-operative MR and the registered images, respectively. In the yellow shadowed circles, corresponding structures are illustrated with dashed yellow circles.

**Figure 4 diagnostics-14-01319-f004:**
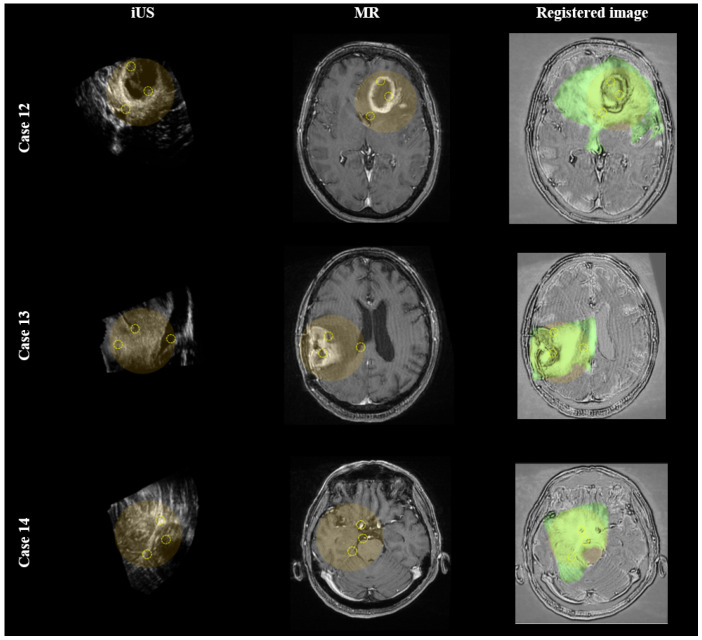
MR to iUS registration results. The first, second, and third rows correspond to cases 12, 13, and 14 of the BITE dataset, respectively. The left, middle, and right columns correspond to the iUS and pre-operative MR and the registered images, respectively. In the yellow shadowed circles, corresponding structures are illustrated with dashed yellow circles.

**Figure 5 diagnostics-14-01319-f005:**
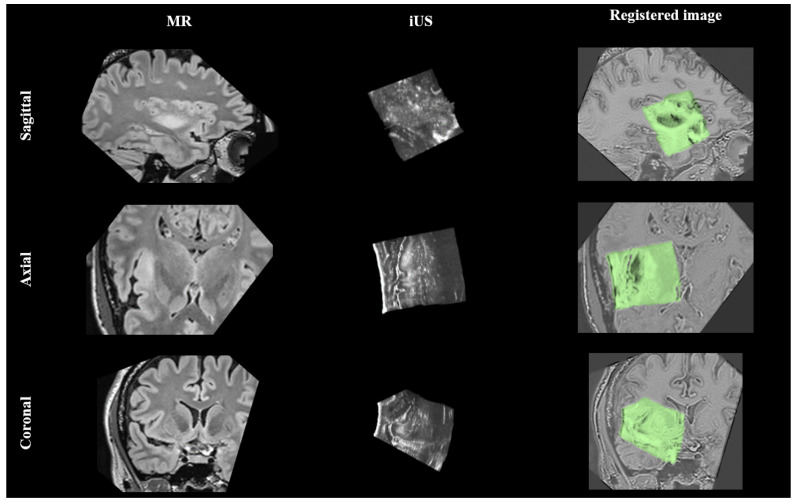
The brain shift indicated in the sagittal, axial, and coronal planes of case 4 in the RESECT dataset. Rows sequentially indicate pre-operative MR, iUS, and registered images.

**Figure 6 diagnostics-14-01319-f006:**
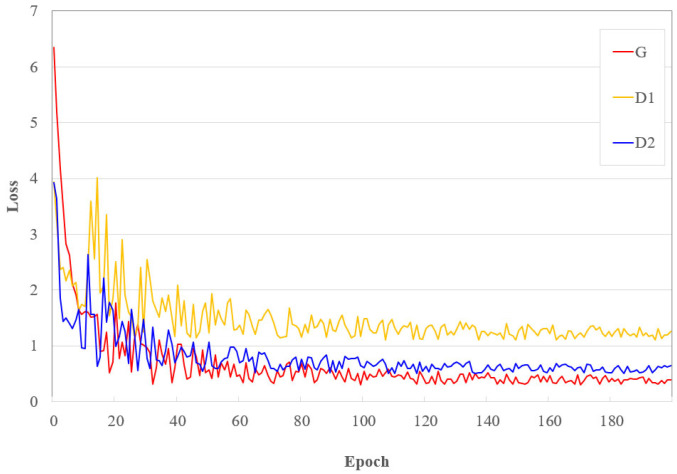
Learning curve for the validation process of the D2BGAN network.

**Table 1 diagnostics-14-01319-t001:** mTRE of D^2^BGAN approach evaluated with corresponding landmarks.

Case	mTRE (mm): Pre-Registration	mTRE (mm): Post-Registration	
		*L*_1_ Loss Function	*L*_2_ Loss Function
#1	1.86	0.8184	0.7377
#2	5.75	1.7625	1.1266
#3	9.63	1.5289	0.9664
#4	2.98	0.8520	0.5183
#5	12.20	1.5311	1.4715
#6	3.34	0.4644	0.2992
#7	1.88	0.7826	0.5380
#8	2.65	0.7109	0.6584
#12	19.76	1.2702	0.8793
#13	4.71	0.59665	0.5946
#14	3.03	0.5196	0.4340
#15	3.37	1.0292	0.8440
#16	3.41	1.1579	1.1256
#17	6.41	0.6660	0.8418
#18	3.66	1.2277	0.7604
#19	3.16	1.2741	1.0548
#21	4.46	1.5884	0.7350
#23	7.05	0.7250	0.4869
#24	1.13	0.4679	0.3606
#25	10.10	1.0692	0.8131
#26	2.93	0.4731	0.3179
#27	5.86	1.0950	1.0493
Mean ± std	5.4240 ± 4.2901	0.9823 ± 0.3958	0.7551 ± 0.3017
*p*-value		0.000026	0.000016

**Table 2 diagnostics-14-01319-t002:** mTRE of D2BGAN approach evaluated with corresponding landmarks of BITE dataset.

Case	mTRE (mm): Pre-Registration	mTRE (mm): Post-Registration
#1	5.88 ± 2.31	1.84 ± 1.23
#2	6.06 ± 1.61	1.50 ± 0.54
#3	8.91 ± 2.02	1.51 ± 0.26
#4	3.87 ± 1.19	1.76 ± 1.09
#5	2.57 ± 1.61	1.94 ± 0.26
#6	2.24 ± 1.05	1.14 ± 0.31
#7	3.02 ± 1.58	1.45 ± 0.16
#8	3.75 ± 1.97	1.00 ± 0.49
#9	5.05 ± 1.33	0.96 ± 0.42
#10	2.99 ± 1.34	0.96 ± 0.25
#11	1.51 ± 0.73	1.30 ± 0.52
#12	3.68 ± 1.85	1.13 ± 0.23
#13	5.13 ± 2.73	1.37 ± 0.57
#14	3.78 ± 1.23	1.08 ± 0.45
Mean ± std	4.18 ± 1.84	1.35 ± 0.49
*p*-Value		0.000039

**Table 3 diagnostics-14-01319-t003:** mTRE for our proposed methods and the state-of-the-art methods on the RESECT dataset.

Method	mTRE (Mean ± Std)
LC^2^	1.75 ± 0.62 mm
SSC	1.67 ± 0.54 mm
NiftyReg	2.90 ±3.59 mm
cDRAMMS	2.28 ± 0.71 mm
FAX	1.21 ± 0.55 mm
CNN	3.91 ± 0.53 mm
D^2^BGAN	0.75 ± 0.30 mm

## Data Availability

The data, that support the findings of this study are openly available as RESECT dataset, reference number [31] and BITE dataset, reference number [32].

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
