# Peer review of "D2BGAN: Dual Discriminator Bayesian Generative Adversarial Network for Deformable MR–Ultrasound Registration Applied to Brain Shift Compensation"

_diagnostics, 2024, doi:10.3390/diagnostics14131319_

Round 1

Reviewer 1 Report

Comments and Suggestions for Authors

The authors present a novel deformable MRI-IUS registration approach to compensate for effects of brain shift in neurosurgical applications, overcoming the challenge of spatial inaccuracy of navigation. The paper is well structured, provides sufficient detail (especially the methods) and is overall well-written.  However, there are some aspects that need to be addressed:

The introduction provides sufficient clinical introduction motivating the need for deformable registration methods of preoperative and intraoperative data. However, overall navigation accuracy is a multifactorial issue. The authors should also cover a bit of the other aspects of accuracy (imaging, technical, patient registration procedure), which is the basis for a more or less “good” initial registration of MRI and iUS data used as starting point for the registration approach.

The authors mention also MRI (CT would also be an option), whereas iUS most effective, fast and so on. What would be the upside of iMRI which is a solid tool for navigation updates (just to complete the view on opportunities for navigation updates)? Are there any other option to compensate for brain shift?

In Section 2.3 the authors state that 264 paired slices were extracted. How was “pairing” performed (initial registration given by the system, which might not be a “good match” in all cases), why only 2D not 3D as I imagine we are talking about a 3D MRI-iUS registration? The same accounts for Section 2.4. Was there any kind of supervision of the approaches performance?

Table 1. Do the case numbers correspond to the numbers in the RESECT data base (non-consecutive numbering), or were there some cases excluded? If, yes, why?

Were any statistical tests regarding TRE performed comparing the different methods?

How is the “bad” preregistration explained (here’s an option to link to the other factors contributing to overall navigation accuracy) (see Results and Discussion)

Minor aspects:

Minor language editing required (e.g. consistent spelling, blanks)

Within the methods section, line 149, “generator G” (G is missing)

“iUS” instead of “US” when intraoperative US is meant

“Figure X” instead of “Fig. X”

Comments on the Quality of English Language

Minor language editing necessary

Author Response

Dear Editor

We are pleased to submit the revised version of our manuscript titled "D2BGAN: Dual Discriminator Bayesian Generative Adversarial Network for Deformable MR-Ultrasound Registration Applied to Brain Shift Compensation." We would like to express our gratitude to the editor for overseeing the review process, and we extend our sincere appreciation to the reviewers for their insightful and constructive feedback, which has significantly contributed to enhancing the quality of our manuscript.

In the following, we describe our response to the comments of the reviewers and the changes we have made to the paper. To help the legibility of the remainder of this response letter, all the reviewer's comments and questions are typeset in bold font. Our responses and remarks are written in plain font. The changes made to our revised manuscript are written in bold blue font. All changes to the original document are highlighted. We think that these changes significantly improve this manuscript. We hope that you now find the manuscript suitable for publication in the journal.

Sincerely yours,

Parastoo Farnia, Ph.D.

Assistant Professor of Biomedical Engineering,

Tehran University of Medical Sciences (TUMS),

Research Center of Biomedical Technology and Robotics (RCBTR).

Reviewer 1:

The authors present a novel deformable MRI-IUS registration approach to compensate for the effects of brain shift in neurosurgical applications, overcoming the challenge of spatial inaccuracy of navigation. The paper is well structured, provides sufficient detail (especially the methods), and is overall well-written.  However, there are some aspects that need to be addressed:

Response: We appreciate the reviewer's insightful comments and will incorporate these improvements to enhance the manuscript.

  1. The introduction provides sufficient clinical introduction motivating the need for deformable registration methods of preoperative and intraoperative data. However, overall navigation accuracy is a multifactorial issue. The authors should also cover a bit of the other aspects of accuracy (imaging, technical, patient registration procedure), which is the basis for a more or less “good” initial registration of MRI and iUS data used as starting point for the registration approach.

Response: We acknowledge that multiple factors, including imaging quality, technical setup, and patient registration procedures, influence navigation accuracy. These factors contribute significantly to the initial registration accuracy of MRI and iUS data, which forms the basis for our proposed registration approach.

Firstly, the quality of preoperative imaging, including resolution and minimal artifacts, forms the foundation for effective registration. Secondly, technical aspects like instrument calibration and tracking device precision directly impact navigation accuracy. Additionally, patient-specific factors such as anatomical variations and intraoperative tissue shifts necessitate continuous updates to maintain alignment. Moreover, the registration procedure itself, involving landmark selection and algorithm choice, significantly influences the precision of navigation. Addressing these multifaceted considerations holistically is paramount to improving the overall accuracy of image-guided navigation systems. Furthermore, user training and experience are essential for optimal system utilization and error minimization. Meanwhile, advanced algorithms capable of deformable registration and real-time adjustment to intraoperative changes in anatomy can enhance navigation precision.

Even if the accuracy of neuro-navigation is optimal, it is restricted by brain shift, which refers to the deformation and displacement of brain tissue during surgery. This phenomenon can result from factors such as the loss of cerebrospinal fluid, changes in intracranial pressure, patient positioning, and surgical manipulation. As the brain tissue moves, the pre-operative imaging used to guide the surgery becomes less accurate, potentially leading to challenges in precisely targeting and avoiding critical areas. To mitigate the effects of brain shift, one common way is to update pre-operative MRI using intraoperative US. This method provides real-time updates on the brain's condition, enhancing the accuracy and safety of neurosurgical procedures.

Therefore, this part is added to the first paragraph of the introduction, page 2:

Navigation accuracy is influenced by multiple factors, including pre-operative imaging quality and resolution, technical aspects like instrument calibration and tracking device precision, and the accuracy of registration algorithms. It is while, the accuracy of neuro-navigation systems would also be limited by brain shift, which refers to the deformation and displacement of the brain tissue during the operation [4].

  1. The authors mention also MRI (CT would also be an option), whereas iUS most effective, fast and so on. What would be the upside of iMRI which is a solid tool for navigation updates (just to complete the view on opportunities for navigation updates)? Are there any other option to compensate for brain shift?

Response: We agree that discussing other modalities and methods for navigation updates would provide a more comprehensive view. As we mention in the manuscript, there are methods and technologies available to address brain shift, such as:

  1. Intraoperative MRI (iMRI): iMRI provides high-resolution images that are beneficial for detailed anatomical structures; however, it is limited by its high cost, operational complexity, prolonged surgery duration, and the need for MRI-compatible instruments.
  2. Intraoperative Ultrasound (iUS): iUS provides real-time imaging and is more portable and cost-effective than iMRI. It can be used to visualize brain structures during surgery and update the navigation system. However, it may not provide the same level of detail as iMRI.
  3. Intraoperative CT (iCT): provides real-time 3D imaging but involves radiation exposure.
  4. Intraoperative X-ray: particularly usefull in procedures requiring bone visualization, though it offers limited soft tissue contrast and involves radiation exposure.
  5. Intraoperative fluorescence imaging: its effectiveness can be limited by the depth of penetration and the specific properties of the dyes used.

Among these modalities, ultrasound is highly regarded for its real-time imaging capabilities, availability, cost-effectiveness, and non-ionizing nature. Intra-operative ultrasound (iUS) is considered a promising imaging method for the delineation of tumor margins and normal brain tissue.

We have now included more information about it in the new version of the manuscript in the second paragraph of the introduction, page 2:

Meanwhile, intraoperative imaging modalities such as X-ray, computed tomography (CT), magnetic resonance imaging (MRI), fluorescence imaging, and ultrasound (US) could provide accurate anatomical localization during the surgery and have the potential to compensate for brain shift. However, each of these methods has its well-known advantages and limitations. Radiation exposure and low spatial resolution in CT, the requirement for an expensive equipped MR-compatible operating room, and the time-consuming nature of MRI, limited imaging depth in fluorescence imaging, are the major challenges of the common intra-operative imaging modalities [6].

Among these modalities, ultrasound is highly regarded for its real-time imaging capabilities, availability, cost-effectiveness, and non-ionizing nature. Intra-operative ultrasound (iUS) is considered a promising imaging method for the delineation of tumor margins and normal brain tissue [7].

  1. In Section 2.3 the authors state that 264 paired slices were extracted. How was “pairing” performed (initial registration given by the system, which might not be a “good match” in all cases), why only 2D not 3D as I imagine we are talking about a 3D MRI-iUS registration? The same accounts for Section 2.4. Was there any kind of supervision of the approaches performance?

Response: You have raised an important point here. As we mention in Section 2.3 “Dataset” training data are 2D-corresponding slices of iUS and pre-operative MR images extracted from the RESECT dataset that are fed to the network without any labels.

Retrospectively, corresponding anatomical landmarks were identified across iUS images of different surgical stages, and between MRI and US, and can be used to validate image registration algorithms. Corresponding anatomical landmarks has helped us to extract paired slices from iUS and MRI images. The quality of landmark identification was assessed with intra- and inter-rater variability [1].

Since the network was trained with 2D slices, the number of data points used for network training was 264 corresponding pairs extracted from the 3D volume of MR and iUS, and according to the unsupervised procedure, no label was used in the training procedure.

This method has enabled effective 2D unsupervised pre-operative MRI to iUS registration, thus contributing to more reliable image registration algorithms and enhancing the overall precision of neurosurgical procedures. In our future work, we are focusing on developing 3D registration techniques to further improve the alignment and accuracy of intraoperative imaging. This advancement aims to provide even more precise navigation and better accommodate the complexities of brain shift during surgery.

  1. Table 1. Do the case numbers correspond to the numbers in the RESECT data base (non-consecutive numbering), or were there some cases excluded? If, yes, why?

Response: Thank you for your attention. All cases from the RESECT dataset [1] are used in this work without any exclusion. As shown in Table 1, 22 cases named from 1 to 27 have been used. The reason for the absence of some cases in Table 1 is that in the mentioned dataset, cases 9, 10, 11, 20, and 22 have not been publicly available, and other researches [2, 3] that has been done in this field has used the same 22 cases used in this manuscript, which include all publicly available cases.

  1. Were any statistical tests regarding TRE performed comparing the different methods?

Response: Thanks for your insightful comment. The p-value is a statistical measure used to determine the significance of results obtained in hypothesis testing. A smaller p-value indicates stronger evidence against the null hypothesis, suggesting that the observed data is unlikely to have occurred by random chance alone. Typically, a p-value threshold (such as 0.05) is set to decide whether to reject the null hypothesis; if the p-value is below this threshold, the results are considered statistically significant. In the context of Table 1, the extremely low p-values (0.000026 for the L1 loss function and 0.000016 for the L2 loss function) indicate a highly significant reduction in mTRE values of post-registration, confirming that the improvements are not due to random variation and highlighting the effectiveness of the D2BGAN approach.

We have now included more information about it in the new version of the manuscript in the section 3 “results”, page 7:

The evaluation of the D2BGAN registration method on the RESECT dataset demonstrates a significant improvement in the registration accuracy. The initial error, which was measured at 5.42 mm, is reduced to 0.75 mm after applying the proposed method, with a statistically significant difference (p-value = 0.000016). This indicates a clear advantage of the D2BGAN registration method in terms of reducing the registration error on the RESECT dataset.

To compare the measurement of the TRE for the L1 and L2 loss functions, we analyze how the mTRE performs throughout the registration. In the Table 1, the p-values (0.000026 for the L1 and 0.000016 for the L2) indicate a highly significant reduction in mTRE values, confirming that the improvements are not due to random variation and highlighting the effectiveness of the D2BGAN approach.

  1. How is the “bad” preregistration explained (here’s an option to link to the other factors contributing to overall navigation accuracy) (see Results and Discussion)

Response:

The value calculated as the mTRE of pre-registration, is influenced by various factors including the size and location of craniotomy, the patient's head position, the biological shift amount, and the location from which the physician performs the iUS imaging. As observed in Table 1, the pre-registration value can vary from 1.13 to 19.76 mm depending on the mentioned factors which can significantly affect the accuracy of registration. One of the strong points of the proposed algorithm is compensating for the occurred shift and reducing it to the range of 0.51 to 1.76, indicating the algorithm's ability to compensate for the occurred biological shift which would be different for each case.

We have now included more information about it in the new version of the manuscript in part "discussion" on page 11, as follows:

The pre-registration accuracy is influenced by various factors including the size and location of craniotomy, the patient's head position, the biological shift amount, and the location from which the physician performs the iUS imaging. As observed in Table 1, the pre-registration value can vary from 1.13 to 19.76 mm depending on the mentioned factors which can significantly affect the accuracy of registration. One of the strong points of the proposed algorithm is compensating for the occurred shift and reducing it to the range of 0.51 to 1.76, indicating the algorithm's ability to compensate for the occurred biological shift which would be different for each case.

Minor aspects:

Minor language editing required (e.g. consistent spelling, blanks)

Response: Thank you for your attention, we have made the correction.

Within the methods section, line 149, “generator G” (G is missing)

Response: Thank you for your comment, we have made the correction.

“iUS” instead of “US” when intraoperative US is meant

Response: We appreciate your attention, we have made the correction.

“Figure X” instead of “Fig. X”

Response: Thank you for your attention, we have made the correction.

References

  1. Xiao, Y., et al., RE troSpective Evaluation of Cerebral Tumors (RESECT): A clinical database of pre‐operative MRI and intra‐operative ultrasound in low‐grade glioma surgeries. Medical physics, 2017. 44(7): p. 3875-3882.
  2. Rivaz, H. and D.L. Collins, Deformable registration of preoperative MR, pre-resection ultrasound, and post-resection ultrasound images of neurosurgery. International journal of computer assisted radiology and surgery, 2015. 10: p. 1017-1028.
  3. Zeineldin, R.A., et al. Towards automated correction of brain shift using deep deformable magnetic resonance imaging-intraoperative ultrasound (MRI-iUS) registration. in Current directions in biomedical engineering. 2020. De Gruyter.

Reviewer 2 Report

Comments and Suggestions for Authors

Deformable MRI-Ultrasound registration is a field in which many researchers are making various attempts. This study attempted to solve this problem using GAN. The authors' attempt is a new approach in this field and has academic value.

Major concern:
Please show specific reasons for using two discriminators. The two classifiers are not responsible for any special functions. I expected that the two classifiers proposed by the authors would share special functions, but was disappointed when they did not.

When training a GAN, it is very difficult to match the training ratio of the classifier and generator. The method proposed by the authors may be able to train the classifier probabilistically properly, but it appears that it will become more difficult to match the training weight.

It should be shown how each training component was optimized and what the learning curve was like during the training process.

Minor conerns:
In the future, generative artificial intelligence such as GAN and diffusion will develop further. Introduce papers explaining the various functions of GAN and diffusion generation technology that will become a future trend, and discuss future development directions. For example, “A feasibility study on the adoption of a generative denoising diffusion model for the synthesis of fundus photographs using a small dataset, Discover Applied Sciences, 2024”

Author Response

Dear Editor

We are pleased to submit the revised version of our manuscript titled "D2BGAN: Dual Discriminator Bayesian Generative Adversarial Network for Deformable MR-Ultrasound Registration Applied to Brain Shift Compensation." We would like to express our gratitude to the editor for overseeing the review process, and we extend our sincere appreciation to the reviewers for their insightful and constructive feedback, which has significantly contributed to enhancing the quality of our manuscript.

In the following, we describe our response to the comments of the reviewers and the changes we have made to the paper. To help the legibility of the remainder of this response letter, all the reviewer's comments and questions are typeset in bold font. Our responses and remarks are written in plain font. The changes made to our revised manuscript are written in bold blue font. All changes to the original document are highlighted. We think that these changes significantly improve this manuscript. We hope that you now find the manuscript suitable for publication in the journal.

Sincerely yours,

Parastoo Farnia, Ph.D.

Assistant Professor of Biomedical Engineering,

Tehran University of Medical Sciences (TUMS),

Research Center of Biomedical Technology and Robotics (RCBTR).

Reviewer 2:

Deformable MRI-Ultrasound registration is a field in which many researchers are making various attempts. This study attempted to solve this problem using GAN. The authors' attempt is a new approach in this field and has academic value.

Response: We appreciate you taking the time to review our paper and provide valuable comments. It was your insightful comments that led to possible improvements in the current version. We have carefully considered the comments and tried our best to address every one of them.

Major concern:

  1. Please show specific reasons for using two discriminators. The two classifiers are not responsible for any special functions. I expected that the two classifiers proposed by the authors would share special functions, but was disappointed when they did not.

Response: One of the most important reasons that two discriminators were used is that we had two images (iUS and MRI) with completely different natures. Our initial goal was that the two discriminators share information, but in the training process of the network, we realized that the shared information does not help to train the network due to the different nature of the images and just increases the number of network parameters.

On the other hand, we investigated the usage of one discriminator instead of two, it also created a heavy task for the discriminator, and the convergence of the network was practically not possible. But by separating the tasks of identifying and diagnosing iUS images and MRI images, convergence was established in the training process.

We have now included more information about it in the new version of the manuscript in Section 2.2 "D2BGAN framework" on page 3, as follows:

In this network, two discriminators were used because of the completely different natures of iUS and MRI images.

And in Section 4 "discussion" on page 11, as follows:

Two discriminators were used in our study to accommodate the diverse nature of intraoperative iUS and MRI images. Initially aimed to share information, this approach proved ineffective, increasing complexity without aiding convergence. Assigning separate tasks to each discriminator improved convergence, effectively managing the distinct characteristics of each modality.

  1. When training a GAN, it is very difficult to match the training ratio of the classifier and generator. The method proposed by the authors may be able to train the classifier probabilistically properly, but it appears that it will become more difficult to match the training weight.

Response: As mentioned in the manuscript (D2BGAN framework section), in a dual discriminator structure, it is crucial to maintain a balance between the discriminators while considering the adversarial relationship with the generator. The strength or weakness of each discriminator affects the overall efficiency of the training process. To achieve balance, the network design and training procedures ensure that both discriminators have the same architecture.

  1. It should be shown how each training component was optimized and what the learning curve was like during the training process.

Response: The entire network is trained with a learning rate of 1×10−5 exponentially decaying to 0.85 of the original value after each epoch, and the batch size is set as 24. Optimization of networks is done based on trial and error, and after training the network to the most optimal mode so that, the balance between the generator and the discriminators is established.

Figure 1 indicates the learning curves for the generator and both discriminators. These curves will illustrate the progression of the loss functions over time, highlighting how each component's performance evolves throughout the training process. As shown in the figure 1, the convergence process of network D1 is slower than network D2 due to the more complex nature of iUS images.

Considering that the journal has a limit for the number of figures in the manuscript, we could not include this figure in the manuscript.

Figure 1. Learning curve of the validation process of the D2BGAN network

We have now included more information about it in the new version of the manuscript in Section 3 "Results" on page 10, as follows:

On the other hand, the results improvement is also due to the loss function used in the network training process. For the G we used  and  and  for D1 and D2 discriminators, respectively. Figure 1 indicates the learning curves for validation loss of the generator and both discriminators. The curve illustrates the progression of the loss functions over time, highlighting how each component's performance evolves throughout the training process. As shown in Figure 6, the convergence process of network D1 is slower than network D2 due to the more complex nature of iUS images. These findings highlight the effectiveness of our proposed D2BGAN method in achieving highly accurate MR-iUS registration on the RESECT dataset.

Figure 6. Learning curve of the validation process of the D2BGAN network

Minor conerns:

  1. In the future, generative artificial intelligence such as GAN and diffusion will develop further. Introduce papers explaining the various functions of GAN and diffusion generation technology that will become a future trend, and discuss future development directions. For example, “A feasibility study on the adoption of a generative denoising diffusion model for the synthesis of fundus photographs using a small dataset, Discover Applied Sciences, 2024”

Response: Generative artificial intelligence technologies such as GANs and diffusion models are rapidly evolving, showing great potential across various domains [1-3]. Future generative AI models will increasingly integrate multiple types of data, such as text, images, audio, and video, to create more sophisticated and contextually aware outputs. These multimodal models will enable applications like generating realistic videos from textual descriptions or creating complex interactive environments for virtual and augmented reality. Generative AI will evolve to become better collaborators with humans, providing creative tools that augment human abilities rather than replace them. This includes more intuitive interfaces, a better understanding of human intentions, and the ability to work seamlessly with human creators in fields like art, music, and design. As generative AI becomes more powerful, there will be a greater emphasis on ethical considerations. This involves developing frameworks and guidelines to ensure AI is used responsibly, addressing issues such as bias, misinformation, and the potential for misuse. Transparency in AI processes and results will also be a key focus.

The future of generative AI is poised to bring about significant advancements and changes across various sectors. The key will be to harness its potential while addressing the associated ethical, technical, and societal challenges. As the technology matures, it will open up new frontiers of creativity, efficiency, and innovation. We have now included more information about it in the new version of the manuscript in section 2.1 "Generative Adversarial Network (GAN)" on page 3, as follows:

Generative artificial intelligence technologies such as GANs and diffusion models are rapidly evolving, showing great potential across various domain [27-29].

References

  1. Radford, A., L. Metz, and S. Chintala, Unsupervised representation learning with deep convolutional generative adversarial networks. arXiv preprint arXiv:1511.06434, 2015.
  2. Yi, X., E. Walia, and P. Babyn, Generative adversarial network in medical imaging: A review. Medical image analysis, 2019. 58: p. 101552.
  3. Kim, H.K., et al., A feasibility study on the adoption of a generative denoising diffusion model for the synthesis of fundus photographs using a small dataset. Discover Applied Sciences, 2024. 6(4): p. 188.
